# Association between Preoperative Psychiatric Morbidities and Mortality after Oncologic Surgery: A Nationwide Cohort Study from 2002 to 2019 in South Korea

**DOI:** 10.3390/jpm13071069

**Published:** 2023-06-29

**Authors:** Tak-Kyu Oh, Hye-Yoon Park, In-Ae Song

**Affiliations:** 1Department of Anesthesiology and Pain Medicine, Seoul National University Bundang Hospital, Seongnam 13620, Republic of Korea; 2Department of Anesthesiology and Pain Medicine, College of Medicine, Seoul National University, Seoul 03080, Republic of Korea; 3Department of Psychiatry, Seoul National University Hospital, Seoul 03080, Republic of Korea; 4Department of Psychiatry, College of Medicine, Seoul National University, Seoul 03080, Republic of Korea

**Keywords:** depression, cancer, mortality, psychiatric morbidities

## Abstract

We aimed to examine whether preoperative psychiatric morbidities affect 30-day postoperative mortality. Using a nationwide registration database in South Korea, the study included all patients who underwent curative cancer surgery from 1 January 2002 to 31 December 2019. Patients underwent surgery for breast, laryngeal, lung, thyroid, gastric, colorectal, esophageal, liver, pancreatic, kidney, bladder, testicular, prostate, vulvar, uterine, or brain cancer. Depression, anxiety disorder, substance abuse, and post-traumatic stress disorder were considered preoperative psychiatric morbidities. Among the 944,794 patients in the final analysis, 5490 (0.6%) died within 30 days of the surgery, and 24,370 (2.6%) had preoperative psychiatric morbidities. Multivariable logistic regression analysis showed that preoperative psychiatric morbidities were associated with a higher (adjusted odds ratio [aOR]: 1.23; 95% confidence interval [CI]: 1.09, 1.39; *p* = 0.001) 30-day mortality rate than the rate noted for patients without preoperative psychiatric morbidities. This association was significant in the breast (aOR: 3.31, 95% CI: 1.36, 8.07; *p* = 0.009), lung (aOR: 1.54, 95% CI: 1.19, 2.01; *p* = 0.001), and kidney (aOR: 1.87, 95% CI: 1.06, 3.31; *p* = 0.03) cancer groups in the subgroup analyses. In South Korea, preoperative psychiatric morbidities are considered to be associated with increased 30-day postoperative mortality.

## 1. Introduction

Cancer is the leading cause of death worldwide, and the global cancer burden is projected to continue to increase in the future [1,2,3]. In the pase, the most common cancer treatment has been curative surgery [4]. In 2015, there were 15.2 million new cases of cancer worldwide, of which more than 80% required surgery [5]. Thus, delivering safe, affordable, and timely cancer surgery is an important health issue for global and national cancer control.

Patients with cancer often suffer from psychiatric distress and are diagnosed with mental illness, which affects their care [6]. In a population-based cohort study conducted in Australia, a 30% higher cancer fatality rate was observed among psychiatric patients than among other participants; however, the incidence was not greater than that in the general population [7]. Although psychiatric morbidities in patients with cancer have been the focus of previous studies [6,7,8,9], the impact of psychiatric illness on surgical outcomes in patients who undergo cancer surgery has not yet been identified. Wikman et al. [10] reported that psychiatric morbidities were associated with poorer survival outcomes after surgery for esophageal cancer in a Swedish cohort. However, information on the relationship between preoperative psychiatric morbidities and postoperative mortality is lacking.

Therefore, we aimed to examine whether preoperative psychiatric morbidities affected 30-day postoperative mortality for various cancer types among patients registered in a nationwide database in South Korea. We hypothesized that preoperative psychiatric morbidities would increase the 30-day postoperative mortality rate.

## 2. Materials and Methods

### 2.1. Study Design and Ethical Statement

We followed the Strengthening the Reporting of Observational Studies in Epidemiology guidelines in this population-based cohort study. The study protocol was approved by the Institutional Review Board of Seoul National University Bundang Hospital (X-2111-721-901) and the big data center of the National Health Insurance Service (NHIS) (NHIS-2021-1-616). The requirement for informed consent was waived because data analyses were performed retrospectively, using anonymized data derived from the South Korean NHIS database.

### 2.2. Study Population

The inclusion criteria in this study were as follows: all patients admitted to the hospital in South Korea who received curative cancer surgery from 1 January 2002 to 31 December 2019. The types of cancer and their codes in the database were as follows: breast (C50), laryngeal (C32), lung (C34), thyroid (C73), gastric (C16), colorectal (C18), esophageal (C15), liver (C22), pancreatic (C25), kidney (C64), bladder (C67), testicular (C62), prostate (C61), vulvar (C51), uterine (C55), and brain cancer (C71). We excluded patients with missing age and sex data.

### 2.3. Preoperative Psychiatric Morbidity (Exposure Variable)

ICD-10 codes for depression (F32, F33, and F34.1), anxiety disorder (F41), substance abuse (F10-19), and post-traumatic stress disorder (PTSD) (F43.1) were used to determine the time of diagnosis of each disease. Preoperative psychiatric morbidities were defined as the occurrence of one of the above ICD codes within 1 year before the date of cancer surgery lisated in the NHIS database. Patients with preoperative psychiatric morbidities were included in the preoperative-PY group, while the others were included in the no-PY group.

### 2.4. Study Endpoint

The primary endpoint of this study was 30-day mortality, defined as mortality within 30 days of cancer surgery.

### 2.5. Covariates

Data regarding age and sex were collected as physical characteristics. All patients were classified according to the following age groups: 1–19, 20–39, 40–59, 60–79, and ≥80 years. As indicators of socioeconomic status (SES), patients’ type of residence and household income level at surgery were collected. Household income level is registered in the NHIS database to determine a patient’s insurance premium. However, individuals who consistently do not pay their insurance premiums or have difficulty supporting themselves financially are included in the Medical Aid Program, and as such, the government covers almost all medical expenses to reduce the patient’s financial burden. We divided all patients, except those in the Medical Aid Program, into four quartiles. The type of residence was classified as urban (Seoul and other metropolitan cities) and rural (all other areas). As an indication of physical comorbidities, the Charlson comorbidity index (CCI) score and information regarding underlying disabilities were collected. The CCI score was calculated using the registered ICD-10 codes, as indicated in Appendix A. All patients were classified into five groups according to the CCI score: 0–2, 3–4, 5–6, 7–8, and ≥9. All individuals with disabilities must be registered in the NHIS database to receive benefits from South Korea’s social welfare system. In this database, patients are divided into six groups according to disability severity. We divided patients into two severity groups: grades 1–3 represented severe disability, and grades 4–6 indicated mild-to-moderate disability.

### 2.6. Statistical Methodology

The clinicopathological characteristics of all patients are presented as numbers, with percentages for categorical variables, and as mean values, with standard deviations (SDs) for continuous variables. We compared the clinicopathological characteristics between the preoperative-PY and no-PY groups via the t-test and Chi-squared test for continuous and categorical variables, respectively. We constructed a multivariable logistic regression model for 30-day postoperative mortality to examine the relationship between preoperative psychiatric morbidity and 30-day mortality. The following covariates were included in the model for multivariable adjustment. Age and sex were included as baseline demographic information. SES-related information, such as household income level, residence, and employment status, was included because a low SES is reportedly associated with increased postoperative mortality risk [11]. The preoperative CCI score and disability at surgery were included to reflect comorbid status because they measure risk in patients undergoing cancer surgery [12]. The types of cancer and year of surgery were also included in the model because they affect postoperative outcomes.

In addition, the preoperative-PY group was divided into four subgroups based on the type of psychiatric morbidity (depression, anxiety disorder, substance abuse, and PTSD), and these groups were included in a separate multivariable model to avoid multicollinearity. The Hosmer–Lemeshow test was used to confirm the goodness-of-fit of the multivariable model. We confirmed no multicollinearity between variables in the multivariable model, as the variance inflation factor was <2.0. Subgroup analyses were also performed according to the type of cancer surgery, age, sex, and preoperative CCI, using multivariable logistic regression modeling. All statistical analyses were performed using IBM SPSS Statistics for Windows, version 25.0 (IBM Corp., Armonk, NY, USA), and statistical significance was set at *p* < 0.05.

## 3. Results

### 3.1. Study Population

The data of 979,868 patients who underwent cancer surgery from 2002 to 2019 were initially screened for use in this study. Among them, 35,074 patients were excluded due to incomplete medical records in the NHIS database regarding age and sex. Thus, 944,794 patients were included in the final analysis. The clinicopathological characteristics of all patients are presented in Table 1. During the 30-day postoperative period, 5490 patients (0.6%) died, and the average hospital stay was 13.4 days (SD: 9.6 days). In total, 24,370 (2.6%) patients exhibited preoperative psychiatric morbidities. More specifically, 13,312 (1.4%), 12,082 (1.3%), 816 (0.1%), and 16 (<0.1%) patients were affected by preoperative depression, anxiety disorder, substance abuse, and PTSD, respectively. The most commonly abused substance was alcohol (723/816, 88.6%). Table 2 summarizes the results comparing clinicopathological characteristics between the preoperative- and no-PY groups. The 30-day mortality rate in the preoperative-PY group was 1.2% (304/24,370), while that in the no-PY group was 0.6% (5186/920,424) (*p* < 0.001).

### 3.2. Results of 30-Day Mortality

Table 3 summarizes the results of the multivariable logistic regression model for 30-day postoperative mortality from 2002 to 2019. In multivariable model 1, the preoperative-PY group had an adjusted odds ratio (aOR) of 1.23 (95% confidence interval [CI]: 1.09, 1.39; *p* = 0.001) for 30-day postoperative mortality compared to the no-PY group. In multivariable model 2, patients affected by preoperative anxiety disorder and substance abuse, respectively, had 1.38-fold (95% CI: 1.18, 1.62; *p* < 0.001) and 2.10-fold (95% CI: 1.34, 3.31; *p* = 0.001) higher odds of 30-day postoperative mortality than those in the no-PY group. However, patients with preoperative depression (*p* = 0.405) and PTSD (*p* > 0.999) did not exhibit a statistically significant difference in 30-day mortality compared with the no-PY group.

### 3.3. Subgroup Analyses

Table 4 presents the results of the subgroup analyses, according to the type of cancer surgery. The preoperative-PY group showed higher odds of 30-day postoperative mortality for breast (aOR: 3.31, 95% CI: 1.36, 8.07; *p* = 0.009), lung (aOR: 1.54, 95% CI: 1.19, 2.01; *p* = 0.001), and kidney (aOR: 1.87, 95% CI: 1.06, 3.31; *p* = 0.031) cancers compared to the no-PY group. Table 5 summarizes the results of the subgroup analyses according to age, sex, and CCI at the time of cancer surgery. Significant associations between preoperative psychiatric morbidity and 30-day postoperative mortality were observed in the 40–59-year-old group (aOR: 1.66, 95% CI: 1.23, 2.24; *p* = 0.001) and the ≥80-year-old group (aOR: 1.26, 95% CI: 1.08, 1.47; *p* = 0.004). Moreover, among men, the preoperative-PY group had 1.32-fold (aOR: 1.32, 95% CI: 1.15, 1.52; *p* < 0.001) higher odds of 30-day postoperative mortality than did the no-PY group.

## 4. Discussion

In this population-based cohort study, preoperative psychiatric morbidity was associated with increased 30-day postoperative mortality in South Korea from 2002 through 2019. This association was more evident in patients who underwent breast, lung, and kidney cancer surgeries. Moreover, among psychiatric morbidities, anxiety disorders and substance abuse were significantly associated with increased 30-day postoperative mortality. Finally, among patients aged ≥40 years and among men, preoperative psychiatric morbidities were also significantly associated with increased 30-day postoperative mortality. These results revealed a novel association between preoperative psychiatric morbidities and survival outcomes of various conditions using subgroup analyses.

The prevalence of psychiatric morbidities, such as depression and anxiety disorder, is lower in South Korea than in other countries, such as the United States [13]. As we used ICD-10 codes registered by physicians or psychiatrists to determine psychiatric morbidities, the apparently low prevalence of such morbidities may be partly ascribed to individuals who did not visit outpatient clinics owing to mild symptoms or poor accessibility. Moreover, the psychiatric morbidity diagnosis rate in South Korea is relatively low. For example, in a recent cohort study, the prevalence of depression in South Korea was 1.6%, 2.5%, and 3.1% in 2001, 2006, and 2011, respectively [14].

The association between preoperative psychiatric morbidity and increased 30-day postoperative mortality has several possible explanations. First, psychiatric morbidities, such as depression, may increase the production of proinflammatory cytokines and cause immune dysregulation, thereby increasing the risks of other morbidities and mortality [15]. Psychiatric morbidities reportedly increase mortality among patients with immune-mediated inflammatory diseases [16]. Moreover, inflammatory and immune responses to surgery are known to affect postoperative complications [17]. Second, a high prevalence of medical comorbidities was previously reported in patients with psychiatric morbidities [18,19]. This finding suggests that patients with preoperative morbidities are more physically ill and have a higher risk of 30-day postoperative mortality. As indicated in Table 2, the preoperative-PY group had higher CCI scores (mean: 4.5 [SD: 2.6]) than the no-PY group (mean: 3.3 [SD: 2.1]), indicating that preoperative psychiatric morbidities increased the risk of 30-day postoperative mortality. Third, patient compliance in the postoperative period might also have influenced 30-day mortality, and patients with psychiatric comorbidities are known to show lower compliance with treatment than other patients [20].

Interestingly, among men and patients aged ≥40 years, preoperative psychiatric morbidities were significantly associated with increased 30-day postoperative mortality. Although we could not verify the mechanism, mortality related to psychiatric morbidities, such as depression, was higher in men than in women in a previous meta-analysis [21]. In a South Korean study, the association between depression and increased mortality was also more significant in men than women [22]. Similarly, in studies of patients with psychiatric morbidities, cancer-related mortality was reportedly higher in men than in women [7,23]. Depressive symptoms affected all-cause mortality in a study of middle-aged and older adults in South Korea [24], which is similar to our result that patients aged ≥40 years were at a high risk for increased 30-day postoperative mortality.

In our subgroup analyses, the positive association between preoperative psychiatric morbidities and increased 30-day postoperative mortality was significant in the breast, lung, and kidney cancer groups. Iglay et al. [25] reported that patients with severe psychiatric comorbidities were more likely to be diagnosed with breast cancer, with a 20% increase in breast cancer-specific mortality. In another cohort study, patients with lung cancer and a history of severe psychiatric morbidities were at a high risk of cancer-specific mortality [26]. Our results are particularly noteworthy, as we included only patients who underwent curative cancer surgery. In a previous study, preexisting psychiatric morbidity was associated with increased mortality in patients with lung cancer [27]. Moreover, patients with kidney cancer reportedly experience psychological distress that required psychosocial care [28]. Our study provides crucial information regarding the association between preoperative psychiatric morbidities and 30-day postoperative mortality in patients who underwent surgery for kidney cancer.

This study had several limitations. First, the NHIS lacks important information, such as body mass index, duration of surgery, alcohol consumption, and smoking history, all of which could have affected the results. Second, this study did not consider the tumor, node, and metastatic stages of each cancer. However, 30-day postoperative mortality, rather than mortality over longer periods, has been recommended as an international standard because it reflects the majority of surgery-related deaths, and it is less likely to be influenced by the cancer stage [29]. Moreover, certain psychiatric comorbidities may be associated with frequent doctor visits. These frequent visits can improve the odds of early cancer detection, which is a factor that might have affected the results of this study. Third, as we used registered ICD-10 codes to define psychiatric morbidities before surgery in this study, we include no data from cases without a diagnosis. For example, individuals who did not visit the outpatient clinic because of mild symptoms or poor accessibility to outpatient clinics would not have been captured in this study. Lastly, the retrospective nature of this study might have introduced bias. Moreover, we cannot exclude the possibility of unmeasured confounders, which might have affected our study results.

## 5. Conclusions

In conclusion, preoperative psychiatric morbidity was related to increased 30-day postoperative mortality in our study of a South Korean cohort. This association was most evident in patients with breast, lung, and kidney cancers, as well as in males and patients aged ≥40 years. Our results suggest that psychiatric morbidities before surgery should be considered as risk factors for increased postoperative mortality among cancer patients. Moreover, this study provides a clinical rationale for future studies focused on reducing postoperative mortality risk in patients with preoperative psychiatric morbidities.

## Figures and Tables

**Table 1 jpm-13-01069-t001:** Clinicopathological characteristics of all patients (*n* = 944,794).

Variable	Number (%)	Mean (SD)
Age, years		58.3 (13.6)
1–19	3553 (0.4)	
20–39	76,764 (8.1)	
40–59	414,138 (43.8)	
60–79	406,976 (43.1)	
≥80	43,363 (4.6)	
Sex, male	423,522 (44.8)	
Residence at surgery		
Urban area	314,147 (33.3)	
Rural area	630,647 (66.7)	
Household income level		
Medical aid program	34,775 (3.7)	
Q1	160,047 (16.9)	
Q2	164,963 (17.5)	
Q3	224,805 (23.8)	
Q4	341,344 (36.1)	
Unknown	18,860 (2.0)	
Having a job at surgery	537,838 (56.9)	
Disability at surgery		
Mild to moderate	59,224 (6.3)	
Severe	28,316 (3.0)	
Preoperative CCI, points		3.4 (2.1)
0–2	498,904 (52.8)	
3–4	269,221 (28.5)	
5–6	33,793 (3.6)	
7–8	112,573 (11.9)	
≥9	30,303 (3.2)	
Preoperative psychiatric morbidity	24,370 (2.6)	
Depression	13,312 (1.4)	
Anxiety disorder	12,082 (1.3)	
Substance abuse	816 (0.1)	
PTSD	16 (0.0)	
LOS, days		13.4 (9.6)
30-day postoperative mortality	5490 (0.6)	
Type of cancer		
Breast cancer	201,477 (21.3)	
Laryngeal cancer	1810 (0.2)	
Lung cancer	85,055 (9.0)	
Thyroid cancer	137,639 (14.6)	
Gastric cancer	217,200 (23.0)	
Colorectal cancer	109,774 (11.6)	
Esophageal cancer	13,379 (1.4)	
Liver cancer	69,427 (7.3)	
Pancreatic cancer	14,738 (1.6)	
Kidney cancer	47,907 (5.1)	
Bladder cancer	9250 (1.0)	
Testicular cancer	3565 (0.4)	
Prostate cancer	2441 (0.3)	
Vulvar cancer	924 (0.1)	
Uterine cancer	8773 (0.9)	
Brain cancer	21,435 (2.3)	
Year of surgery		
2002	22,397 (2.4)	
2003	25,531 (2.7)	
2004	28,108 (3.0)	
2005	30,156 (3.2)	
2006	33,471 (3.5)	
2007	36,482 (3.9)	
2008	53,486 (5.7)	
2009	57,517 (6.1)	
2010	60,670 (6.4)	
2011	66,423 (7.0)	
2012	67,448 (7.1)	
2013	67,685 (7.2)	
2014	67,830 (7.2)	
2015	69,185 (7.3)	
2016	72,175 (7.6)	
2017	62,735 (6.6)	
2018	60,217 (6.4)	
2019	63,278 (6.7)	

SD, standard deviation; CCI, Charlson comorbidity index; PTSD, post-traumatic stress disorder; LOS, length of hospital stays.

**Table 2 jpm-13-01069-t002:** Comparison of clinicopathological characteristics between the preoperative- and no-PY groups.

Variable	Preoperative-PY Group *n* = 24,370	No-PY Group *n* = 920,424	*p*-Value
Age, years	62.3 (13.3)	58.2 (13.6)	<0.001
Sex, male	11,015 (45.2)	412,507 (44.8)	0.237
Residence at surgery			0.680
Urban area	8133 (33.4)	306,014 (33.2)	
Rural area	16,237 (66.6)	614,410 (66.8)	
Household income level			<0.001
Medical aid program	2279 (9.4)	32,496 (3.5)	
Q1	4092 (16.8)	155,955 (16.9)	
Q2	4150 (17.0)	160,813 (17.5)	
Q3	5365 (22.0)	219,440 (23.8)	
Q4	8042 (33.0)	333,302 (36.2)	
Unknown	442 (1.8)	18,418 (2.0)	
Having a job at surgery	12,406 (50.9)	525,432 (57.1)	<0.001
Disability at surgery			<0.001
Mild to moderate	2265 (9.3)	56,959 (6.2)	
Severe	1652 (6.8)	26,664 (2.9)	
Preoperative CCI, points	4.5 (2.6)	3.3 (2.1)	<0.001
LOS, days	21.0 (15.0)	13.2 (9.4)	<0.001
30-day mortality	304 (1.2)	5186 (0.6)	<0.001
Type of cancer			
Breast cancer	5774 (23.7)	195,703 (21.3)	<0.001
Laryngeal cancer	125 (0.5)	1685 (0.2)	<0.001
Lung cancer	3399 (13.9)	81,656 (8.9)	<0.001
Thyroid cancer	1428 (5.9)	136,211 (14.8)	<0.001
Gastric cancer	4323 (17.7)	212,877 (23.1)	<0.001
Colorectal cancer	3172 (13.0)	106,602 (11.6)	<0.001
Esophageal cancer	598 (2.5)	12,781 (1.4)	<0.001
Liver cancer	1577 (6.5)	67,850 (7.4)	<0.001
Pancreatic cancer	533 (2.2)	14,205 (1.5)	<0.001
Kidney cancer	1148 (4.7)	46,759 (5.1)	0.009
Bladder cancer	376 (1.5)	8874 (1.0)	<0.001
Testicular cancer	37 (0.2)	3528 (0.4)	<0.001
Prostate cancer	79 (0.3)	2362 (0.3)	0.040
Vulvar cancer	33 (0.1)	891 (0.1)	0.072
Uterine cancer	349 (1.4)	8424 (0.9)	<0.001
Brain cancer	1419 (5.8)	20,016 (2.2)	<0.001

PY, psychiatric morbidity; CCI, Charlson comorbidity index; LOS, length of hospital stays.

**Table 3 jpm-13-01069-t003:** Multivariable logistic regression model for 30-day postoperative mortality from 2002 to 2019.

Variable	aOR (95% CI)	*p*-Value
Preoperative-PY group (vs no-PY group) (model 1)	1.23 (1.09, 1.39)	0.001
Preoperative-PY group, in detail (vs no-PY group) (model 2)		
Depression	0.93 (0.78, 1.11)	0.405
Anxiety disorder	1.38 (1.18, 1.62)	<0.001
Substance abuse	2.10 (1.34, 3.31)	0.001
PTSD	0.00 (0.00-)	0.999
Other covariates in model 1 (below)		
Age, years	1.06 (1.06, 1.07)	<0.001
Sex, male (vs female)	1.58 (1.49, 1.68)	<0.001
Residence at surgery		
Urban area	1	
Rural area	1.12 (1.06, 1.19)	<0.001
Household income level		
Medical aid program	1	
Q1	0.61 (0.55, 0.69)	<0.001
Q2	0.63 (0.56, 0.71)	<0.001
Q3	0.57 (0.51, 0.63)	<0.001
Q4	0.49 (0.44, 0.54)	<0.001
Unknown	0.53 (0.43, 0.66)	<0.001
Having a job at surgery	0.97 (0.92, 1.03)	0.345
Disability at surgery		
Mild to moderate	1.02 (0.93, 1.11)	0.684
Severe	1.53 (1.38, 1.70)	<0.001
Preoperative CCI, points	1.19 (1.18, 1.21)	<0.001
Type of cancer		
Breast cancer	0.07 (0.04, 0.11)	<0.001
Laryngeal cancer	0.39 (0.21, 0.74)	0.004
Lung cancer	1.02 (0.74, 1.41)	0.918
Thyroid cancer	0.12 (0.08, 0.19)	<0.001
Gastric cancer	0.63 (0.46, 0.87)	0.005
Colorectal cancer	1.60 (1.16, 2.20)	0.004
Esophageal cancer	1.67 (1.18, 2.35)	0.004
Liver cancer	1.12 (0.81, 1.55)	0.496
Pancreatic cancer	1.43 (1.01, 2.02)	0.043
Kidney cancer	0.51 (0.36, 0.73)	<0.001
Bladder cancer	0.86 (0.58, 1.26)	0.428
Testicular cancer	0.85 (0.44, 1.64)	0.636
Prostate cancer	0.28 (0.12, 0.62)	0.002
Vulvar cancer	0.00 (0.00-)	0.990
Uterine cancer	1.44 (1.05, 1.98)	0.024
Brain cancer	2.45 (1.76, 3.43)	<0.001
Year of surgery		
2002	1	
2003	0.89 (0.71, 1.11)	0.294
2004	0.86 (0.69, 1.06)	0.156
2005	0.78 (0.63, 0.97)	0.024
2006	0.74 (0.60, 0.92)	0.005
2007	0.90 (0.74, 1.10)	0.314
2008	0.85 (0.70, 1.03)	0.101
2009	0.93 (0.77, 1.12)	0.423
2010	0.82 (0.68, 0.99)	0.039
2011	0.88 (0.73, 1.07)	0.196
2012	0.82 (0.68, 0.99)	0.044
2013	0.67 (0.55, 0.82)	<0.001
2014	0.74 (0.61, 0.89)	0.002
2015	0.72 (0.60, 0.87)	0.001
2016	0.58 (0.48, 0.70)	<0.001
2017	0.61 (0.51, 0.74)	<0.001
2018	0.61 (0.51, 0.74)	<0.001
2019	0.55 (0.46, 0.67)	<0.001

aOR, adjusted odds ratio; CI, confidence interval; PTSD, post-traumatic stress disorder; PY, psychiatric morbidity; CCI, Charlson comorbidity index.

**Table 4 jpm-13-01069-t004:** Subgroup analyses according to the type of cancer surgery.

Variable	aOR (95% CI)	*p*-Value
Breast cancer (*n* = 5774)		
Preoperative-PY group (vs no-PY group)	3.31 (1.36, 8.07)	0.009
Laryngeal cancer (*n* = 125)		
Preoperative-PY group (vs no-PY group)	0.00 (0.00-)	0.996
Lung cancer (*n* = 3399)		
Preoperative-PY group (vs no-PY group)	1.54 (1.19, 2.01)	0.001
Thyroid cancer (*n* = 1428)		
Preoperative-PY group (vs no-PY group)	1.62 (0.37, 7.01)	0.521
Gastric cancer (*n* = 4323)		
Preoperative-PY group (vs no-PY group)	1.24 (0.93, 1.65)	0.151
Colorectal cancer (*n* = 3172)		
Preoperative-PY group (vs no-PY group)	1.12 (0.90, 1.40)	0.303
Esophageal cancer (*n* = 598)		
Preoperative-PY group (vs no-PY group)	0.92 (0.52, 1.64)	0.776
Liver cancer (*n* = 1577)		
Preoperative-PY group (vs no-PY group)	1.40 (0.96, 2.06)	0.081
Pancreatic cancer (*n* = 533)		
Preoperative-PY group (vs no-PY group)	1.31 (0.70, 2.44)	0.402
Kidney cancer (*n* = 1148)		
Preoperative-PY group (vs no-PY group)	1.87 (1.06, 3.31)	0.031
Bladder cancer (*n* = 376)		
Preoperative-PY group (vs no-PY group)	1.39 (0.61, 3.13)	0.432
Testicular cancer (*n* = 37)		
Preoperative-PY group (vs no-PY group)	0.00 (0.00-)	0.997
Prostate cancer (*n* = 79)		
Preoperative-PY group (vs no-PY group)	0.00 (0.00-)	0.996
Vulvar cancer (*n* = 33)		
Preoperative-PY group (vs no-PY group)	0.00 (0.00-)	0.999
Uterine cancer (*n* = 349)		
Preoperative-PY group (vs no-PY group)	1.13 (0.26, 4.93)	0.873
Brain cancer (*n* = 1419)		
Preoperative-PY group (vs no-PY group)	0.82 (0.33, 1.75)	0.605

aOR, adjusted odds ratio; CI, confidence interval; PY, psychiatric morbidity.

**Table 5 jpm-13-01069-t005:** Subgroup analyses according to age, sex, and CCI.

Variable	aOR (95% CI)	*p*-Value
Age: 1–19 years old		
Preoperative-PY group (vs no-PY group)	0.00 (0.00-)	0.997
Age: 20–39 years old		
Preoperative-PY group (vs no-PY group)	0.52 (0.07, 3.85)	0.526
Age: 40–59 years old		
Preoperative-PY group (vs no-PY group)	1.66 (1.23, 2.24)	0.001
Age: ≥80 years old		
Preoperative-PY group (vs no-PY group)	1.26 (1.08, 1.47)	0.004
Sex: Male		
Preoperative-PY group (vs no-PY group)	1.32 (1.15, 1.52)	<0.001
Sex: Female		
Preoperative-PY group (vs no-PY group)	1.09 (0.87, 1.36)	0.468
Preoperative CCI: 0–2		
Preoperative-PY group (vs no-PY group)	1.19 (0.71, 1.98)	0.517
Preoperative CCI: 3–4		
Preoperative-PY group (vs no-PY group)	1.41 (1.17, 1.72)	<0.001
Preoperative CCI: 5–6		
Preoperative-PY group (vs no-PY group)	0.68 (0.41, 1.02)	0.062
Preoperative CCI: 7–8		
Preoperative-PY group (vs no-PY group)	1.62 (1.23, 2.13)	0.001
Preoperative CCI: ≥9		
Preoperative-PY group (vs no-PY group)	0.98 (0.76, 1.26)	0.851

aOR, adjusted odds ratio; CI, confidence interval; PY, psychiatric morbidity; CCI, Charlson comorbidity index.

## Data Availability

Data are available upon reasonable request to the corresponding authors.

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
