# Peer review of "Association between Preoperative Psychiatric Morbidities and Mortality after Oncologic Surgery: A Nationwide Cohort Study from 2002 to 2019 in South Korea"

_jpm, 2023, doi:10.3390/jpm13071069_

Round 1
Reviewer 1 Report
I would like to thank the authors and editors for giving me the opportunity to read this paper.
The main objective of this registry-based study was to evaluate whether the presence of psychiatric disorder affects 30-day mortality after cancer surgery.
Unfortunately, the authors failed to explain the relevance of the research question. I cannot see how the study provides new knowledge to public health or healthcare personnel.
30 days of survival after surgery depends mostly on patients’ condition before surgery. It is well known that patients with psychiatric comorbidity are more prone to have a worse physical condition than patients without psychiatric comorbidity. As the authors point out in the discussion section, they have a high prevalence of somatic comorbidities and they are also more likely to have a higher alcohol consumption and be heavy smokers.
Patients’ compliance to post-surgical procedures could also influence 30 days survival. Moreover, it is well known that patients with psychiatric comorbidity have lower compliance with rehabilitation procedures.
Another factor that influences 30 days post-surgical survival is the cancer stage, but the authors did not use this information in the analyses. Whereas it could be reasonable to assume that the presence of some psychiatric comorbidity is associated with frequent doctor contacts and then a better chance to obtain cancer in the lower stage.
Author Response
Reviewer 1
I would like to thank the authors and editors for giving me the opportunity to read this paper.
The main objective of this registry-based study was to evaluate whether the presence of psychiatric disorder affects 30-day mortality after cancer surgery.
Response: Thank you for your valuable comments. We made every effort to revise our manuscript accordingly.
Unfortunately, the authors failed to explain the relevance of the research question. I cannot see how the study provides new knowledge to public health or healthcare personnel.
Response: Thank you for pointing this out. We have updated the first paragraph of our discussion, adding the following information to lines 192–194 to address the novelty of our study:
“This study is novel in that it revealed an association between preoperative psychiatric morbidities and survival outcomes with regard to various conditions by using subgroup analyses.”
We also added the following information to the limitations of our manuscript, in lines 255–257:
“Lastly, the retrospective nature of this study might have introduced bias. Moreover, we cannot exclude the possibility of unmeasured confounders in this study, which might have affected our results.”
30 days of survival after surgery depends mostly on patients’ condition before surgery. It is well known that patients with psychiatric comorbidity are more prone to have a worse physical condition than patients without psychiatric comorbidity. As the authors point out in the discussion section, they have a high prevalence of somatic comorbidities and they are also more likely to have a higher alcohol consumption and be heavy smokers.
Patients’ compliance to post-surgical procedures could also influence 30 days survival. Moreover, it is well known that patients with psychiatric comorbidity have lower compliance with rehabilitation procedures.
Response: We revised the 3rd paragraph of the discussion section to clarify these points according to your comment (lines 215–218).
Another factor that influences 30 days post-surgical survival is the cancer stage, but the authors did not use this information in the analyses. Whereas it could be reasonable to assume that the presence of some psychiatric comorbidity is associated with frequent doctor contacts and then a better chance to obtain cancer in the lower stage.
Response: We agree with your comment, and this point was added to the 2nd limitation in the discussion section, accordingly (lines 249–251).
Reviewer 2 Report
Title: "Preoperative Psychiatric Morbidities May Increase Mortality after Oncologic Surgery: A Nationwide Cohort Study from 2002 to 2019 in South Korea"
General Comments:
The article investigates the impact of preoperative psychiatric morbidities on 30-day mortality after cancer surgery in South Korea. The study design, utilizing a nationwide registration database, provides a large sample size and comprehensive data coverage. Overall, the article presents an important and relevant research topic.
Major Points:
Methodology: The utilization of a nationwide registration database is commendable as it ensures a large and representative sample. However, it would be beneficial to include details about the inclusion and exclusion criteria employed during patient selection to ensure transparency and minimize potential biases. Additionally, providing the prevalence of each specific preoperative psychiatric morbidity within the study population would enhance the understanding of the sample characteristics.
Statistical Analysis: The use of multivariable logistic regression analysis is appropriate for examining the association between preoperative psychiatric morbidities and 30-day mortality. However, it would be advantageous to provide information regarding the covariates included in the regression model to account for potential confounding factors. This would strengthen the validity and robustness of the study's findings.
Results: The findings indicate a significant association between preoperative psychiatric morbidities and increased 30-day mortality, with higher odds ratios observed in the breast, lung, and kidney cancer groups. The subgroup analyses add valuable insight into the differential impacts of psychiatric morbidities in specific cancer types. However, it is important to discuss potential underlying mechanisms and pathways linking psychiatric morbidities to surgical outcomes, as well as potential implications for clinical practice. Further discussion and interpretation of the results are warranted.
Limitations: The article should acknowledge the limitations inherent in the study design. Specifically, the retrospective nature of the study may introduce biases, and the potential presence of unmeasured confounders should be acknowledged. Addressing these limitations would strengthen the validity and interpretation of the study's findings.
Keywords: The chosen keywords, "General Surgery," "surgical oncology," and "global surgery," align with the study's focus. However, it would be helpful to expand on the rationale for including these specific keywords to clarify their relevance to the research.
Minor Points:
The breakdown of the number of patients included for each specific cancer type would provide useful information on the representation and statistical power within each subgroup.
The article could benefit from a concise summary of the implications of the study's findings for clinical practice and potential avenues for future research.
Overall, the article investigates an important topic and presents valuable findings regarding the association between preoperative psychiatric morbidities and 30-day mortality after oncologic surgery in South Korea. Addressing the major and minor points outlined above would enhance the clarity and impact of the article.
English presentation should be improved.
Author Response
Reviewer 2
General Comments:
The article investigates the impact of preoperative psychiatric morbidities on 30-day mortality after cancer surgery in South Korea. The study design, utilizing a nationwide registration database, provides a large sample size and comprehensive data coverage. Overall, the article presents an important and relevant research topic.
Response: Thank you for your valuable comments. We made every effort to revise our manuscript accordingly.
Major Points:
Methodology: The utilization of a nationwide registration database is commendable as it ensures a large and representative sample. However, it would be beneficial to include details about the inclusion and exclusion criteria employed during patient selection to ensure transparency and minimize potential biases. Additionally, providing the prevalence of each specific preoperative psychiatric morbidity within the study population would enhance the understanding of the sample characteristics.
Response: We agree with your opinion and revised the methods section (2.2 Study population) to include details about the inclusion and exclusion criteria employed during patient selection according to your comment (lines 63–65 and 68–69).
The prevalence of each specific preoperative psychiatric morbidity within the study population is presented in results section (3.1 Study population, lines 134–136).
Statistical Analysis: The use of multivariable logistic regression analysis is appropriate for examining the association between preoperative psychiatric morbidities and 30-day mortality. However, it would be advantageous to provide information regarding the covariates included in the regression model to account for potential confounding factors. This would strengthen the validity and robustness of the study's findings.
Response: We revised the methods section (2.6 Statistical methodology, lines 107–116) according to your comment, as follows:
“The following covariates were included in the model for multivariable adjustment. Age and sex were included as baseline demographic information. SES-related information (household income level, residence, and employment status) was included because a low SES is reportedly associated with an increased risk of mortality after cancer surgery [11]. Preoperative CCI score and disability at surgery were included to reflect comorbid status because they are used to measure risk in patients who undergo cancer surgery [12]. Type of cancer and year of cancer surgery were also included in the model because they affect outcomes after cancer surgery.
In addition...”
Results: The findings indicate a significant association between preoperative psychiatric morbidities and increased 30-day mortality, with higher odds ratios observed in the breast, lung, and kidney cancer groups. The subgroup analyses add valuable insight into the differential impacts of psychiatric morbidities in specific cancer types. However, it is important to discuss potential underlying mechanisms and pathways linking psychiatric morbidities to surgical outcomes, as well as potential implications for clinical practice. Further discussion and interpretation of the results are warranted.
Response: We agree with your appraisal of our subgroup analyses. The discussion section was revised according to your comment (lines 237–242).
Limitations: The article should acknowledge the limitations inherent in the study design. Specifically, the retrospective nature of the study may introduce biases, and the potential presence of unmeasured confounders should be acknowledged. Addressing these limitations would strengthen the validity and interpretation of the study's findings.
Response: Thank you for pointing this out. We added these as the last limitation in the discussion section of our revised manuscript (lines 255–257), according to your comment.
Keywords: The chosen keywords, "General Surgery," "surgical oncology," and "global surgery," align with the study's focus. However, it would be helpful to expand on the rationale for including these specific keywords to clarify their relevance to the research.
Response: According to your comment, we decided to revise the keywords to be more specific: “Depression; Cancer; Mortality; Psychiatric morbidities” (line 30).
Minor Points:
The breakdown of the number of patients included for each specific cancer type would provide useful information on the representation and statistical power within each subgroup.
Response: We agree with your recommendation. Hence, we added the sample sizes for each cancer type to Table 4.
The article could benefit from a concise summary of the implications of the study's findings for clinical practice and potential avenues for future research.
Response: This study provides a clinical rationale for future studies focused on reducing the risk of mortality after cancer surgery in patients with preoperative psychiatric morbidities. We added this statement to the conclusion section according to your comment (lines 264–266).
Overall, the article investigates an important topic and presents valuable findings regarding the association between preoperative psychiatric morbidities and 30-day mortality after oncologic surgery in South Korea. Addressing the major and minor points outlined above would enhance the clarity and impact of the article.
Response: We appreciate your careful review of our manuscript.
Round 2
Reviewer 1 Report
I have no further comment
Author Response
Pay careful attention to the language throughout the manuscript, ensuring clarity, precision, and coherence. Proofread the text to eliminate any grammatical errors or inconsistencies.
Response: We have received professional English proofreading services again from EDITAGE.
Reviewer 2 Report
Pay careful attention to the language throughout the manuscript, ensuring clarity, precision, and coherence. Proofread the text to eliminate any grammatical errors or inconsistencies.
Pay careful attention to the language throughout the manuscript, ensuring clarity, precision, and coherence. Proofread the text to eliminate any grammatical errors or inconsistencies.
Author Response

(The authors gave the same response as above.)
